# Conditional Generation of 3D Brain Tumor ROIs via VQGAN and Temporal-Agnostic Masked Transformer

**Meng Zhou**[1,2]                                                                SIMONZHOU@CS.TORONTO.EDU
**Farzad Khalvati**[1,2,3]                                                     FARZAD.KHALVATI@UTORONTO.CA

[1] *Neurosciences & Mental Health Research Program, SickKids Research Institute, Toronto, ON, Canada*

[2] *Department of Computer Science, University of Toronto, Toronto, ON, Canada*

[3] *Department of Medical Imaging, University of Toronto, Toronto, ON, Canada*

**Editors:** Accepted for publication at MIDL 2024

## Abstract

Neuroradiology studies often suffer from lack of sufficient data to properly train deep learning models. Generative Adversarial Networks (GANs) can mitigate this problem by generating synthetic images to augment training datasets. However, GANs sometimes are unstable and struggle to produce high-fidelity images. An alternative solution is Diffusion Probabilistic Models, but these models require extensive computational resources. Additionally, most of the existing generation models are designed to generate the entire image volumes, rather than the regions of interest (ROIs) such as the tumor region. Research shows that it is easier to classify tumor types based on ROIs than the entire image volumes. To this end, we present a class-conditioned ROI generation framework that combines a vector-quantization GAN and a class-conditioned masked Transformer to generate high-resolution and diverse 3D brain tumor ROIs. We further propose a temporal-agnostic masking strategy to effectively learn relationships between semantic tokens in the latent space. Our experiments demonstrate that the proposed method can generate high-quality 3D MRIs of brain tumor regions for both low- and high-grade glioma (LGG/HGG) in the BraTS 2019 dataset. Using the generated data, our approach demonstrates superior performance compared to several baselines in a downstream task of brain tumor type classification. Our proposed method has the potential to facilitate accurate diagnosis of rare brain tumors using MRI-based machine learning models.

**Keywords:** Generative Adversarial Networks, Transformer, Image Generation, 3D MRI, Data Augmentation

## 1. Introduction

Gliomas are the most frequent primary adult brain tumor types within the central nervous system (Menze et al., 2014; Bakas et al., 2017). Among all variations of gliomas, high-grade glioma (HGG) accounts for the majority of cases, and low-grade glioma (LGG) accounts for less common cases. For both variations, a commonly used technique for diagnosing is the multi-parametric Magnetic Resonance Imaging (MRI) equipped with different sequences such as T1-, T2-weighted, and Fluid Attenuated Inversion Recovery (FLAIR) (Menze et al., 2014). Each modality provides distinct biological information about the tumor, aiding radiologists in determining the tumor type. However, distinguishing between HGG and LGG remains challenging, and misdiagnosis may lead to suboptimal prognoses (Mzoughi et al., 2020).

In recent years, deep learning-based methods have proven to be one of the effective ways for adult brain tumor classification tasks using brain MR images (Ge et al., 2020; Hao et al., 2021; Namdar et al., 2022; Tandel et al., 2020). However, the requirement for large training datasets poses challenges in medical imaging, especially for rare diseases such as LGG, leading to potential overfitting and poor generalization to unseen datasets. Several works aim to mitigate the imbalanced data problem. One line of work is to use the transfer learning approach by pre-training models on huge datasets (i.e., ImageNet), and then fine-tuning them on domain-specific datasets (Ghazal et al., 2022; Tak et al., 2023; Ullah et al., 2022). Another line of work is to synthesize MRIs using Generative Adversarial Network (GAN) (Volokitin et al., 2020; Kwon et al., 2019; Sun et al., 2020; Xia et al., 2020) or Diffusion-based methods (Khader et al., 2022; Peng et al., 2022a; Dorjsembe et al., 2023; Sanchez et al., 2022) to alleviate the need for extensive datasets. However, GANs for image generation can be unstable, produce blurry images, and encounter mode collapse problems (Kwon et al., 2019). As an alternative approach, Diffusion Probabilistic models have been proposed and demonstrated superior performance over GANs (Müller-Franzes et al., 2022), but these methods are extremely computationally expensive when synthesizing full-resolution MRIs, thus posing challenges in both the training and inference stage. More recently, autoregressive transformer models have attracted increasing attention in image generation tasks (Esser et al., 2021a,b; Huang et al., 2023). The key idea behind such models is to obtain discretized feature maps from a Vector-Quantization GAN (VQGAN) model and then use the transformer model to learn the compositions. Autoregressive transformers have been extended to medical images (Pinaya et al., 2023; Tudosiu et al., 2022; Zhou et al., 2023), to unconditionally generate brain MR images. However, one limitation of these models is the lack of conditioning; separate models need to be trained for different tasks, which becomes time-consuming and resource-intensive. Therefore, a condition-based image generation model is important for practical applications in real clinical settings. Moreover, a group of works (Sajjad et al., 2019; Mzoughi et al., 2020; Srinivasan et al., 2023) have demonstrated the effectiveness of using the tumor region of interest (ROI) for classifying tumor types because ROIs contain less information than the whole-image that may negatively affect the results. Hence, this work aims to explore generating different brain MRI tumor ROIs based on their pathology labels. To this end, we introduce the first class-conditional generation framework for synthesizing 3D brain tumor MRI ROIs. Our model is built upon the previous work (Zhou et al., 2023) and extended to a conditional generation paradigm. Our framework has three modules: a 3D-VQGAN image encoder to extract high-level feature maps while concurrently learning the importance score for each region in the feature maps; an Exponential Moving Averages(EMA) codebook with $l_2$-norm lookup for converting feature maps into discrete semantic tokens (Peng et al., 2022b); and a temporal-agnostic masked transformer to learn the relationships between discrete tokens. We evaluated our proposed method in the BraTS 2019 dataset and demonstrated superior performance over several baselines on both image generation quality and the downstream HGG vs. LGG classification task. **Our contributions are as follows: (1).** We propose the first image generation framework for different tumor types based on the given class label. **(2).** We use a *classifier-guidance approach* to learn the importance score for each region in the encoded feature maps. **(3).** We propose a novel *temporal-agnostic hybrid masking strategy* which uses the importance score to mask tokens to prevent any information leakage.

**(4).** Experiments show our proposed method outperforms several baselines in both image generation quality and the downstream classification task.

## 2. Materials and Methods

### 2.1. Model Architecture

We adopted and extended the VQGAN (Esser et al., 2021b) and recently proposed 3D-VQGAN (Zhou et al., 2023) with some modifications detailed below for class-conditional generation of brain tumor ROIs.

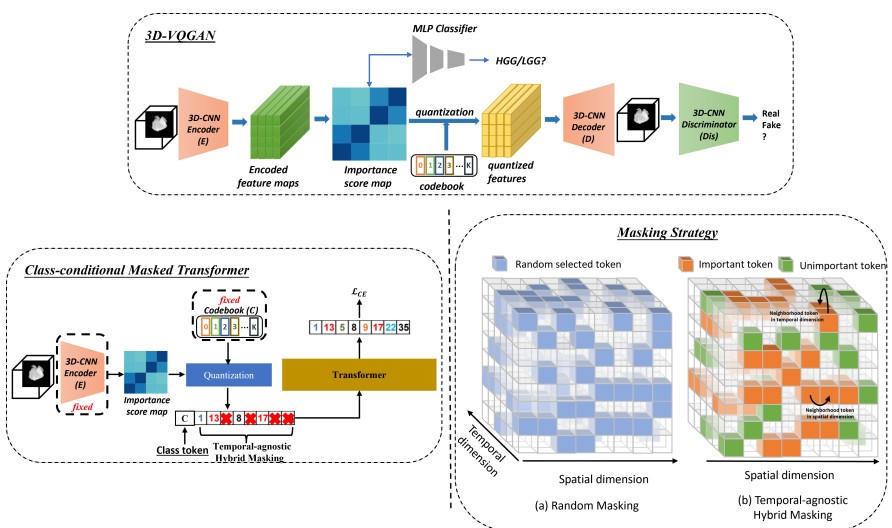

Figure 1: Detailed overview of the proposed method. Our method contains two modules, *Top:* a 3D-VQGAN model to encode 3D inputs, generate importance score for each region, and further quantize to discrete tokens. *Bottom left:* a class-conditional masked transformer to capture the long-term dependency via mask token modeling based on the importance score and class label information. *Bottom right:* difference between random and our proposed masking strategy.

**Stage 1. 3D-VQGAN:** The first stage is shown at the top of Figure 1. We train all modules presented to learn efficient data representation through a reconstruction task in this stage. Our encoder, decoder, and discriminator follow the same design as in (Zhou et al., 2023), except we replace batch normalization with group normalization to stabilize the training process for small batch sizes (Wu and He, 2018).

**Importance Score Map:** We use a lightweight scoring network $f$ before quantization to assign an importance score for each region in encoded feature maps. Let the encoded map with size $z_e \in \mathbb{R}^{H \times W \times Dp \times n_z}$ where $H, W, Dp, n_z$ denote the height, width, depth, and the number of feature maps, respectively. Then, for each region $r_i \in \mathbb{R}^{n_z}$, its score is defined by $s_i = f(r_i)$, where $i = 1, ..., H \times W \times Dp$. The larger the score $s_i$ is, the more important the feature region $r_i$ is. To learn $f$, we use a *classifier-guidance* approach by using an auxiliary MLP-based classifier $f_{cls}$ after $f$ to classify tumor types (e.g., HGG vs. LGG). We hypothesize that regions with the higher scores (i.e., important regions) are the key to

differentiating between two tumor types, and thus by optimizing both $f$ and $f_{cls}$, the model can identify important feature regions that are specific to either LGG or HGG tumors.

**Quantization:** In the quantization step, the latent feature maps are quantized by replacing each one with its closest corresponding vector in codebook $C$. Formally, we train a learnable codebook $C = \{c_i\}_{i=1}^{K}$ that transforms feature vectors $z_e$ to $K := H \times W \times Dp$ discrete tokens $c_q, q \in [1, K]$ by the nearest neighbor search in $C$, and each token $c_q$ includes an embedding vector $c_z \in \mathbb{R}^{n_z}$. We use the $l_2$ normalization for codebook lookup, as done in (Yu et al., 2021). Finally, we stack $K$ quantized feature vectors back to the original latent shape and feed it into the decoder $D$ to produce reconstructed images.

**Stage 2. Class-conditional Masked Transformer:** In this stage, we propose a novel **temporal-agnostic hybrid masking** strategy based on importance scores computed in Stage 1, which is inspired by DropBlock (Ghiasi et al., 2018) and BERT (Devlin et al., 2018). The 3D images are initially represented in the latent space, and the encoder $E$, decoder $D$, and codebook $C$ are fixed, with only the transformer being trained. The encoded feature map $z_e$ of size $H \times W \times Dp \times n_z$ is quantized into a set of $L$ discrete tokens, where $L = H \times W \times Dp$. We first set the masking ratio $\alpha$ and randomly sample $N = L \times \alpha$ tokens to be masked. These tokens are then divided into two equal subsets: $N_1$ and $N_2$, which denote the number of important tokens and unimportant tokens to be masked based on their importance score. Let $\mathbf{Y} = \{y_i\}_{i=1}^{L}$ be the raster-scan linearized discrete tokens and each of $y_i$ associates with their importance score $s_i$. The importance score and the corresponding tokens are sorted in descending order, denoted as $\mathbf{Y}'$. For $N_1$ important tokens, we randomly sample $\lceil \frac{N_1}{2} \rceil$ tokens from the ***top-k = 25%*** of $\mathbf{Y}'$. For each selected token, we also mask along with their spatial or temporal neighborhood tokens. Unlike random masking, this *blockwise masking* around each selected token prevents information leakage from neighbors, enhancing the model's learning ability on important tokens and preventing short-cut learning. The special [MASK] token is used to mask out these important tokens and their associated blocks. For $N_2$ unimportant tokens, we randomly sample them from the remaining $(1 - k) \times \mathbf{Y}'$ tokens and replace them with $N_2$ randomly selected tokens from the codebook $C$. Importantly, we ensure that $N_1$ and $N_2$ are non-overlapping ($N_1 \cap N_2 = \emptyset$). We denote $\mathbf{M} = \{m_i\}_{i=1}^{L}$ be the mask for each of the discrete tokens, where $m_i = 1$ if the token $i$ is *unmasked* and $m_i = 0$ if the token $i$ is *masked out*. Finally, we prepend a class label indicating HGG or LGG sample at the start of each indices sequence. During training, the objective is to reconstruct the masked tokens using unmasked ones. The introduced noise from masked-out tokens is hypothesized to enhance the transformer model's ability to learn relationships between semantic tokens, improving overall robustness. The proposed masking strategy and its difference between random masking is depicted in the bottom right of Figure 1.

**Classification:** For the downstream classification task between LGG and HGG tumor types, we use a standard 3D ResNet-50 model (Hara et al., 2017) that takes 3D tumor ROIs as inputs and outputs two class probabilities for two tumor types.

## 2.2. Loss Function

We employ the same loss function as done in (Zhou et al., 2023). We use a combination of the pixel differences loss ($\mathcal{L}_{pixel}$), perceptual loss ($\mathcal{L}_{perp}$) (Johnson et al., 2016), GAN-based feature matching loss ($\mathcal{L}_{match}$) (Ge et al., 2022), 3D image gradient loss ($\mathcal{L}_{grad}$), codebook

loss ($\mathcal{L}_{codebook}$) (Esser et al., 2021b), and the discriminator loss ($\mathcal{L}_{Dis}$) in the first stage. See Equation (1) and Equation (2) for details.

$$\mathcal{L}_{pixel} = \|x - \hat{x}\|_1, \ \mathcal{L}_{perp} = \sum_{j=1}^{6} \|f^i(x_j) - f^i(\hat{x_j})\|_2^2, \ \mathcal{L}_{match} = \|f^i_{Dis}(x) - f^i_{Dis}(\hat{x})\|_1, \quad (1)$$

$$\mathcal{L}_{grad} = \|\nabla(A(x)) - \nabla(A(\hat{x}))\|_2^2 + \|\nabla(R(x)) - \nabla(R(\hat{x}))\|_2^2 + \|\nabla(S(x)) - \nabla(S(\hat{x}))\|_2^2$$

$$\mathcal{L}_{Dis} = \mathbb{E}_{x \sim p_d}[max(0, 1 - D(x))] + \mathbb{E}_{\hat{x} \sim p_{\hat{d}}}[max(0, 1 + D(\hat{x})],$$
$$\mathcal{L}_{codebook} = \|sg[E(x)] - c_z\|_2^2 + \beta\|sg[c_z] - E(x)\|_2^2 \quad (2)$$

Where $x$ is the original image and $\hat{x}$ is the reconstructed image, $\nabla(\cdot)$ computes the $x$- and $y$-direction gradients of the image, $A(x), R(x), S(x)$ represents slicing over Axial, Coronal and Sagittal plane, respectively. Additionally, we use the standard cross-entropy loss $\mathcal{L}_{ce}$ between class logits and class labels for our auxiliary classifier $f_{cls}$. Aggregating all the loss terms together yields the loss objective in Equation (3) for the first stage of the framework:

$$\min_{E,D,C}(\max_{Dis}(\mathcal{L}_{Dis}))$$
$$\min_{E,D,C} C_1 * (\lambda_1\mathcal{L}_{pixel} + \lambda_2\mathcal{L}_{perp} + \lambda_3\mathcal{L}_{match} + \lambda_4\mathcal{L}_{grad} + \lambda_5\mathcal{L}_{codebook}) + C_2 * \mathcal{L}_{ce} \quad (3)$$

Where $\lambda_i, i \in [1,5]$ is the weighting factor between different loss terms. We follow previous publications (Ge et al., 2022; Khader et al., 2022) to set $\lambda_1 = \lambda_3 = 4$ and $\lambda_2 = \lambda_5 = 1$. We also set $\lambda_4 = 4$ and $\beta$ in $\mathcal{L}_{codebook}$ to be 1. $C_1$ and $C_2$ are balancing factors between the main task and auxiliary task, we empirically set $C_1 = 0.8$ and $C_2 = 0.2$.

For the transformer model, we use the cross entropy loss between the reconstructed token sequence and the ground truth token sequence as shown in Equation (4) to optimize the transformer.

$$\mathcal{L}_{transformer} = -\mathbb{E}_{\mathbf{Y} \in \mathcal{D}}(\sum_{\forall i, m_i=0} logp(y_i|\mathbf{Y}_M)) \quad (4)$$

Where $\mathcal{D}$ is the training dataset, $\mathbf{Y}_M$ denotes the *unmasked* tokens, thus the masked tokens can conditioned on these unmasked tokens during training.

### 2.3. Data and Preprocessing

We used the FLAIR sequence data from the BraTS 2019 dataset (Bakas et al., 2017, 2018; Menze et al., 2014). The data contains 259 HGG patients and 76 LGG patients. We used this dataset since it is the latest version of the dataset which provides labels for the brain tumor pathology classification, i.e., HGG vs. LGG. We reshaped the data from $240 \times 240 \times 155$ to $128 \times 128 \times 128$ and normalized all pixel intensities in $[-1, 1]$. To achieve this, we first remove all zero-valued slices in both the brain images and the segmentations, since we are interested in the slices with the brain tumor present. Then, we obtain the ROIs by multiplying the images with masks. Finally, we center-crop the region based on the segmentation mask to a target size of $128 \times 128 \times 128$.

## 3. Experiments

For the first stage of the proposed 3D-VQGAN-cond model, we train for 10k epochs with an initial learning rate of 0.0001 and cosine decay to 0 for all sub-modules, a mini-batch size of 3, and with the Adam optimizer (Kingma and Ba, 2014). We set the codebook size $K = 1024$. For the second stage, we train the transformer for 5k epochs using a learning rate of $4.5e - 06$, a mini-batch size of 3, and the AdamW optimizer (Loshchilov and Hutter, 2017). We set the mask ratio $\alpha = 0.5$. We randomly held out 25 patients from both HGG and LGG as a standalone test set. The rest of the data is used to train our model.

To assess the usability of our generated data, we conducted two sets of classification experiments: **Experiment (1).** We aimed to determine if models trained on synthetic data are better than those trained without or only using a portion of synthetic data. Following the approach in (Zhou et al., 2023), we compared our classification model pre-trained with *both synthetic LGG and HGG* to one pre-trained with *real HGG and synthetic/real LGG*. We use the same amount of data for pre-training and the **same data** for fine-tuning. **Experiment (2).** Our goal was to investigate whether increasing the number of synthetic data for pre-training enhances classification performance. We generated 250 synthetic HGG and LGG from baseline models and our proposed model to pre-train the classification model. The data for fine-tuning remained consistent. More details can be found in Appendix A. The ablations on the *top-k* ratio can be found in Appendix B due to page limit.

**Baseline Model & Comparison.** For comparison of image generation results, we consider five state-of-the-art methods, 3D-WGAN-GP (Gulrajani et al., 2017), 3D-$\alpha$WGAN (Kwon et al., 2019), 3D-Med-DDPM (Dorjsembe et al., 2023), Medical Diffusion (Khader et al., 2022) and 3D-VQGAN (Zhou et al., 2023). We re-implemented the Medical Diffusion to make it class-conditioned (Medical Diffusion-C) which can be jointly trained on LGG and HGG data, as our class-conditioned baseline. Other baselines were rerun *separately* for two data classes. For classification, we establish a baseline where we only use traditional augmentations, ensuring a fair comparison with other methods. We evaluate the quality of generated MRI ROIs using three commonly used metrics in previous publications (Kwon et al., 2019; Peng et al., 2022a; Dorjsembe et al., 2023; Zhou et al., 2023): maximum mean discrepancy (MMD) (Gretton et al., 2012), multi-slice structure similarity (MS-SSIM) (Rosca et al., 2017), and the Fréchet Inception Distance (FID) (Heusel et al., 2017). FID score is computed in three views (Axial, Coronal, Sagittal) to reflect the nature of medical images. The classification performance is evaluated using AUC, F1-Score, and Accuracy. **Generating Synthetic MRI Data.** To generate 3D tumor ROIs, we start with a class token, either 0 for LGG or 1 for HGG, and then have the transformer model predict and complete the rest of the indices. Next, we obtain the embedding vectors of each index from the codebook $C$ and then feed them into decoder $D$ to produce the final images. For other baselines, we follow the exact procedure as stated in their paper.

## 4. Results and Discussions

### 4.1. Results for Generated Images

In Figure 2, we compare LGG and HGG ROIs generated by baseline models and our proposed method. The center three slices in the Axial plane are shown for better visual quality.

We observed that both 3D-WGAN-GP and 3D-$\alpha$WGAN produce images that lack detail and exhibit major artifacts. 3D-Med-DDPM and Medical Diffusion/-C introduce noise and checkerboard artifacts. Generated images from both 3D-VQGAN and our proposed method contain the detailed attributes of the tumor and exhibit high image fidelity. Quantitative metrics are computed over 250 generated HGG and LGG samples as shown in Table 1. It can be seen that our proposed method performs best in terms of preserving diversity based on the MS-SSIM score. For the MMD score on LGG data, our method outperforms all methods except 3D-$\alpha$WGAN, which we argue that this is not a fair comparison because it exhibits a severe mode collapse problem (99.4 in MS-SSIM). For the MMD score on HGG data, our method is slightly worse than Medical Diffusion/-C, but it still outperforms other baselines. For FID, our method consistently outperforms on FID-A score, and the other two FID-S and FID-C scores are very close to the best performance. It is also worth noting that all baselines except Medical Diffusion-C are trained *separately*, whereas ours and Medical Diffusion-C are trained *jointly* on both LGG and HGG data. Our performance 1) outperforms Medical Diffusion-C in most of the quantitative metrics, and 2) exceeds or is on par with other baselines indicating our method can effectively learn and distinguish between two tumor types and significantly reduce the time needed to train separate models. More visualizations can be found in Appendix C.

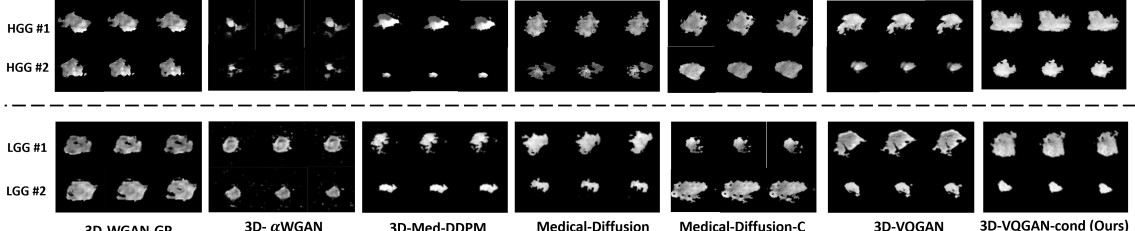

Figure 2: Qualitative comparison between generated and real LGG and HGG ROIs. We show the center three consecutive slices in the Axial plane for each ROI sample. Zoom in for a better view.

## 4.2. Classification Results

We trained a classification model to validate the efficacy of the proposed 3D-VQGAN-cond model in distinguishing between HGG and LGG brain tumor types. Table 2 shows the classification results for Experiments (1) and (2) from Section 3. For Experiment (1), we showed that the model pre-trained with both synthetic HGG and LGG data outperforms all baselines, including the model trained with traditional augmentations and trained only on synthetic LGG data. This highlights the substantial improvement in classification performance achieved through pre-training with synthetic data, alleviating the demand for extensive real data to train effective classification models. For Experiment (2), we demonstrated that the classification performance improves when the number of synthetic data used for pre-training increases. These results collectively indicate that synthetic ROI data can be effectively used to pre-train deep models and only a small amount is needed for fine-tuning. In addition, the improvements from both experiments are statistically significant (two-sided t-test $p < 0.05$) compared to the previous SOTA (3D-VQGAN), which further

Table 1: Quantitative results of class-conditioned generation. Values in '()' are the absolute difference to the real MS-SSIM score (85.3 for LGG, 88.6 for HGG).

| Method | MMD ($10^4$) ↓ | MS-SSIM (%) | FID-A ↓ | FID-C ↓ | FID-S ↓ |
|---|---|---|---|---|---|
| LGG Results | | | | | |
| 3D-WGAN-GP | 1.98 | 93.4 (8.1) | 65.9 | 55.4 | 45.5 |
| 3D-$\alpha$WGAN | 1.61 | 99.4 (14.1) | 79.3 | 69.1 | 73.6 |
| 3D-Med-DDPM | 1.83 | 93.3 (8.0) | 62.6 | 46.5 | 43.3 |
| Medical Diffusion | 1.78 | 92.9 (7.6) | 31.6 | **30.5** | 37.1 |
| Medical Diffusion-C | **1.72** | 89.8 (4.5) | 26.2 | 32.6 | 37.2 |
| 3D-VQGAN | 1.79 | 92.7 (7.4) | 24.1 | 31.4 | 36.1 |
| 3D-VQGAN-cond | **1.72** | **87.9 (2.6)** | **23.7** | 32.9 | **35.2** |
| HGG Results | | | | | |
| 3D-WGAN-GP | 2.44 | 97.5 (8.9) | 53.5 | 50.2 | 50.6 |
| 3D-$\alpha$WGAN | 2.34 | 98.9 (10.3) | 122.3 | 145.7 | 153.1 |
| 3D-Med-DDPM | 2.50 | 95.9 (7.3) | 84.6 | 61.4 | 58.6 |
| Medical Diffusion | **1.41** | 89.4 (0.8) | 30.0 | 26.3 | 23.2 |
| Medical Diffusion-C | 1.44 | 90.7 (2.1) | 32.3 | 27.2 | **20.5** |
| 3D-VQGAN | 1.63 | 90.6 (2.0) | 32.1 | 29.3 | 31.4 |
| 3D-VQGAN-cond | 1.57 | **88.5 (0.1)** | **29.1** | **24.4** | 26.3 |

validate that both LGG and HGG samples generated by our proposed 3D-VQGAN-cond model have better image quality and fidelity compared to other baselines. More results can be found in Appendix D.

Table 2: Results for all experiments as described in Section 3. We run all for three trials and report as mean±standard deviation. Trad. Aug. is the short for traditional augmentations.

(a) Experiment (1)

| Method | AUC | F1-Score | Accuracy |
|---|---|---|---|
| Trad. Aug. | 0.66±0.03 | 0.63±0.03 | 0.59±0.03 |
| 3D-WGAN-GP | 0.64±0.08 | 0.62±0.05 | 0.56±0.01 |
| 3D-$\alpha$WGAN | 0.70±0.09 | 0.59±0.09 | 0.58±0.06 |
| 3D-Med-DDPM | 0.69±0.03 | 0.64±0.01 | 0.61±0.06 |
| Medical Diffusion | 0.71±0.09 | 0.62±0.07 | 0.59±0.02 |
| Medical Diffusion-C | 0.66±0.03 | 0.65±0.02 | 0.58±0.08 |
| 3D-VQGAN | 0.72±0.03 | 0.67±0.02 | 0.65±0.04 |
| Ours | **0.77±0.03**[*] (**p=0.02**) | **0.71±0.02**[*] (**p=0.03**) | **0.67±0.04** |

(b) Experiment (2)

| Method | AUC | F1-Score | Accuracy |
|---|---|---|---|
| 3D-$\alpha$WGAN | 0.71±0.04 | 0.65±0.02 | 0.64±0.06 |
| 3D-Med-DDPM | 0.71±0.09 | 0.69±0.02 | 0.65±0.03 |
| Medical Diffusion | 0.73±0.04 | 0.67±0.02 | 0.61±0.02 |
| Medical Diffusion-C | 0.75±0.03 | 0.68±0.02 | 0.66±0.03 |
| 3D-VQGAN | 0.78±0.04 | 0.71±0.06 | 0.70±0.06 |
| Ours | **0.80±0.02**[*] (**p=0.04**) | **0.74±0.02**[*] (**p=0.03**) | **0.70±0.01** |

## 5. Conclusions

We propose the first class-conditional generation framework for LGG and HGG brain tumor types based on VQGAN and masked Transformer. The conditional scheme enables generating different types of tumors in a unified framework, rather than in separate models that require a large amount of time and resources to train. Our proposed method performs better or on par with several baseline models in image quality metrics such as MS-SSIM, slice-wise FID, and MMD. Using the generated data, our method yields the best classification performance compared to all other baselines.

## Acknowledgments

This research has been made possible with the financial support of the Canadian Institutes of Health Research (CIHR) (Funding Reference Number: 184015). The authors would like to thank Landy Xu for her suggestions on the figures.

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

## Appendix A. Training Details

### A.1. Implementation Details of the Scoring Network

The scoring network $f$ is learned using an auxiliary MLP, $f_{cls}$, through the classifier-guidance approach we described in Section 2.1. To learn $f$, we first use the encoder $E$ to get the latent feature map $z_e \in \mathbb{R}^{H \times W \times Dp \times n_z}$, then for each region $r_i \in \mathbb{R}^{n_z}$, its score is defined by $s_i = f(r_i)$, where $i = 1, ..., H \times W \times Dp$. Next, we normalize the encoded feature map, $z_e^{norm} = LayerNorm(z_e)$ and further multiply with the predicted importance score as modulating factors, $z_e^{'} = z_e^{norm} * \mathbf{S}$ where $\mathbf{S}$ is the set of importance scores ($\{s_i\}, i = 1, ..., H \times W \times Dp$) obtained from $f$, then $f_{cls}$ takes $z_e^{'}$ as the input and output class probabilities.

### A.2. Model Training

All programs were implemented in Pytorch, and all models were trained on a single TESLA V100 GPU. Additionally, we applied the automatic mixed precision in the PyTorch library during the training process (Subramaniam et al., 2022). The overall training time for our 3D-VQGAN-cond model takes about 7 GPU days to complete. For our transformer architecture, we used the same one in (Esser et al., 2021b). Our code is available at here.

For classification, we pre-trained for 50 epochs with a batch size of 8 and a learning rate of 0.001 with Adam optimizer for all models using synthetic data. These models are then finetuned using a batch size of 10, and a learning rate of 0.01. For models that do not involve the use of synthetic data (i.e., traditional augmentation), we trained those with a batch size of 8 and a learning rate of 0.01 with Adam optimizer. All classification models are optimized by focal loss (Lin et al., 2017), as we noticed that it is better than the standard cross entropy loss.

**Data Split.** As discussed in the main text, we randomly hold out 25 HGG and LGG patients (50 in total) as the standalone test data. For the rest of the 234 HGG patients and 51 LGG patients, we design two sets of training data combinations for traditional and non-traditional augmentation methods. For the traditional augmentation baseline, we augment the LGG data by rotating 30 degrees, scaling by 1.5 times larger, left-right flipping, and elastic deformation to form a balanced dataset of 234 cases for both HGG and LGG. For all other non-traditional augmentation models, we pre-trained on 183 real HGGs and 183 synthetic LGGs, and then fine-tuned with 51 real HGGs and 51 real LGGs for **Experiment (1)** in Section 3 follows the setup in (Zhou et al., 2023); we pre-trained on 250 synthetic HGGs and LGGs, and then fine-tuned with 51 real HGGs and 51 real LGGs for **Experiment (2)**. We ensure that the data for fine-tuning is the same across experiments for a fair comparison. Furthermore, during the fine-tuning stage in both experiments, we used 85% of the data for optimizing the model and the remaining 15% of the data for validation, and we repeated this process three times to ensure the robustness of our model. We also ensure that there is no overlap between the validation data in the three runs.

## Appendix B. Ablation Study

### B.1. Ablation on top-k ratio

Recall that the top-$k$ ratio acts as the ratio of important tokens of all discrete semantic tokens obtained from encoded feature maps (e.g., 512 tokens in this work). We argue that the ratio controls the **trade-off between the diversity of generated images and their quality**, i.e., how close is the synthetic to real distribution. If the top-$k$ ratio is small, i.e., $k = 0$, then we treat every token as the unimportant token, and the proposed masking strategy will *degrade to random masking*. Random masking poses an issue of information leakage, impeding the transformer model's ability to effectively learn relationships between regions, especially those pertinent to tumors. Therefore, generated images may lack diversity in critical regions, but the overall distribution may be close to the real one. Hence, the ablation study is conducted on altering the value of $k$ for our proposed 3D-VQGAN-cond model, we compared $k = 0\%$, $k = 25\%$, and $k = 50\%$ and computed the MS-SSIM score to evaluate the diversity and MMD score to evaluate the distance between distributions, as shown in Table 3. We fix the overall masking ratio $\alpha = 0.5$ for all ablations. When $k = 0\%$ (random masking), the MMD score is low but it has a slightly higher MS-SSIM score. For $k = 25\%$, our method exhibits a slightly higher MMD score compared with $k = 0\%$ but has a significantly lower MS-SSIM score (note that the MS-SSIM score is computed over 1000 randomly selected pairs). When $k = 50\%$, we observed that there is a dramatic performance degradation in the MMD score and the MS-SSIM score seems to have minimal change compared to $k = 25\%$. To balance this trade-off, we select $k = 25\%$ in our study. We believe block-wise masking on important tokens helps the transformer model better learn the relationship between other tokens, increasing the possibility of generating diverse images based on important regions.

Table 3: Ablation study on the *top-k* ratio used in our masked transformer model. Values in '()' are the absolute difference to the real MS-SSIM score (85.3 for LGG, 88.6 for HGG).

| | LGG | | HGG | |
|---|---|---|---|---|
| *top-k* ratio | MMD ($10^4$) ↓ | MS-SSIM | MMD($10^4$) ↓ | MS-SSIM |
| 0% | 1.67 | 89.4 (4.1) | 1.46 | 89.7 (1.1) |
| 25% | 1.72 | 87.9 (2.6) | 1.57 | 88.5 (0.1) |
| 50% | 2.14 | 87.3 (2.0) | 1.80 | 89.2 (0.6) |

### B.2. Ablation on number of neighbor tokens

We have also conducted ablations on the number of neighbor tokens to be masked around important tokens, as described in Section 2.1. In our work, we selected only one neighborhood token (we denote as single-side) in either spatial or temporal dimension to be masked, given that our latent feature maps are relatively compact ($8 \times 8 \times 8$), and masking more tokens around the important one may lose too much feature information for the model to

be learned and reconstructed effectively during training. In the ablation, we fix the mask ratio $\alpha = 0.5$, top-k ratio $k = 25\%$, and mask *two neighborhood tokens*, i.e., for a given important token, we mask two more tokens to its left and right in the spatial dimension, or successor and predecessor tokens in the temporal dimension (we denote as double-sided). The quantitative results are reported in Table 4, we observed that masking more tokens led to performance degradation, which validates our claim above.

Table 4: Ablation study on the number of neighbor tokens in our masked transformer model. Values in '()' are the absolute difference to the real MS-SSIM score (85.3 for LGG, 88.6 for HGG).

|  | LGG | | HGG | |
|---|---|---|---|---|
|  | MMD $(10^4) \downarrow$ | MS-SSIM | MMD$(10^4) \downarrow$ | MS-SSIM |
| single-side | 1.72 | 87.9 (2.6) | 1.57 | 88.5 (0.1) |
| double-sided | 2.07 | 88.9 (3.6) | 1.91 | 89.9 (1.3) |

## Appendix C.  More on Generated Images

In this section, we provide more visualizations of the generated LGG and HGG from our proposed method and other baseline methods. The additional visualization of generated LGG samples from all methods is shown in Figure 3; additional HGG samples visualization is shown in Figure 4.

We observe that all GAN-based baselines produce images with noise and blurry edges, and the image quality is low. For the 3D-Med-DDPM, the intensity range in the generated samples seems to mismatch the real samples, and it looks unreal compared with the real ROIs. For Medical Diffusion/-C, the generated images suffer from minor checkerboard artifacts (visible when zoomed in). 3D-VQGAN sometimes produces blurry images (sample 2 in HGG), but overall, the generated images are smooth and do not have any checkerboard artifacts or noises. Finally, for our method, the images exhibit high-resolution with no noise, no blurry edges, and no checkerboard artifacts, the contrast inside the generated ROIs looks very similar to the real ROIs.

We also provide a visualization of the importance score map learned by the proposed scoring network $f$ in Section 2.1. Visualizing the importance map provides insight into how the model has learned each region in the latent feature map, as depicted in Figure 5. Lighter regions indicate higher importance. The score map is obtained by interpolating the original importance score in the latent space $(8 \times 8 \times 8)$ to the original image size and overlay with the original image. We can observe that our model can effectively identify important regions within the tumor, and provide a reliable reference for the masked transformer model.

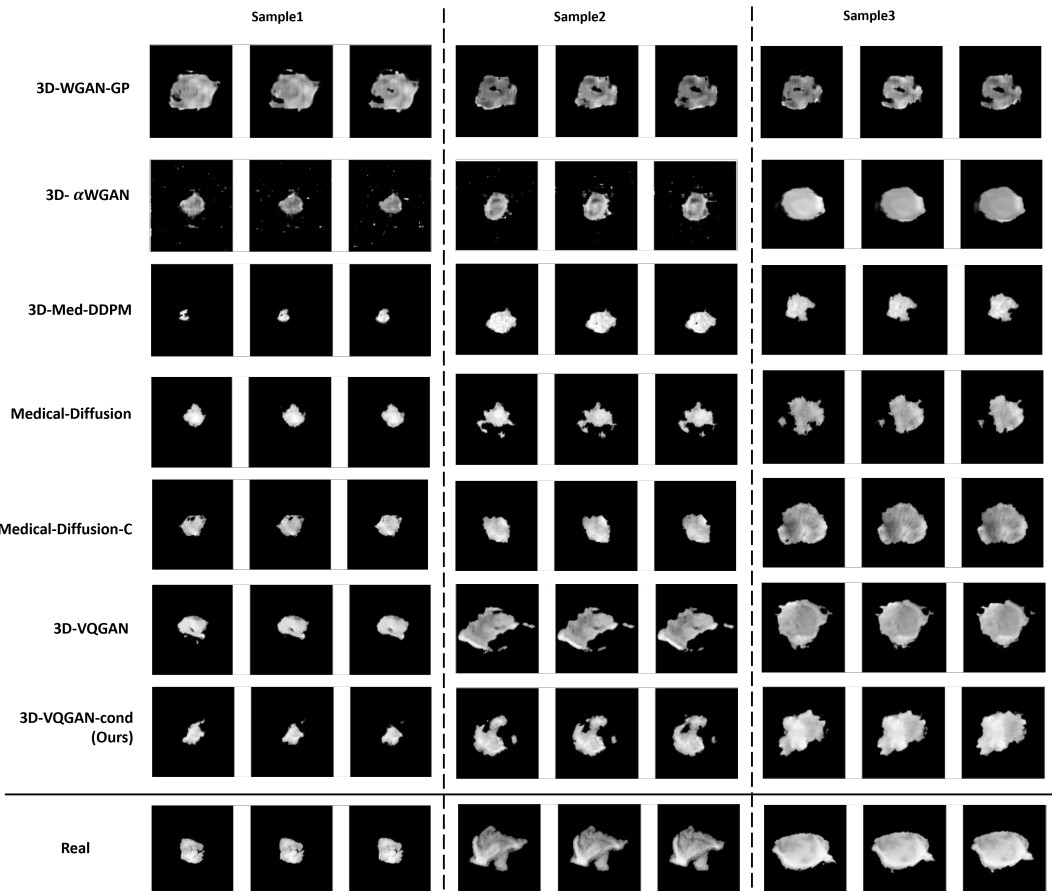

Figure 3: Additional generated samples of LGG data. We show the center three consecutive slices in the Axial plane for each ROI sample. Zoom in for a better view.

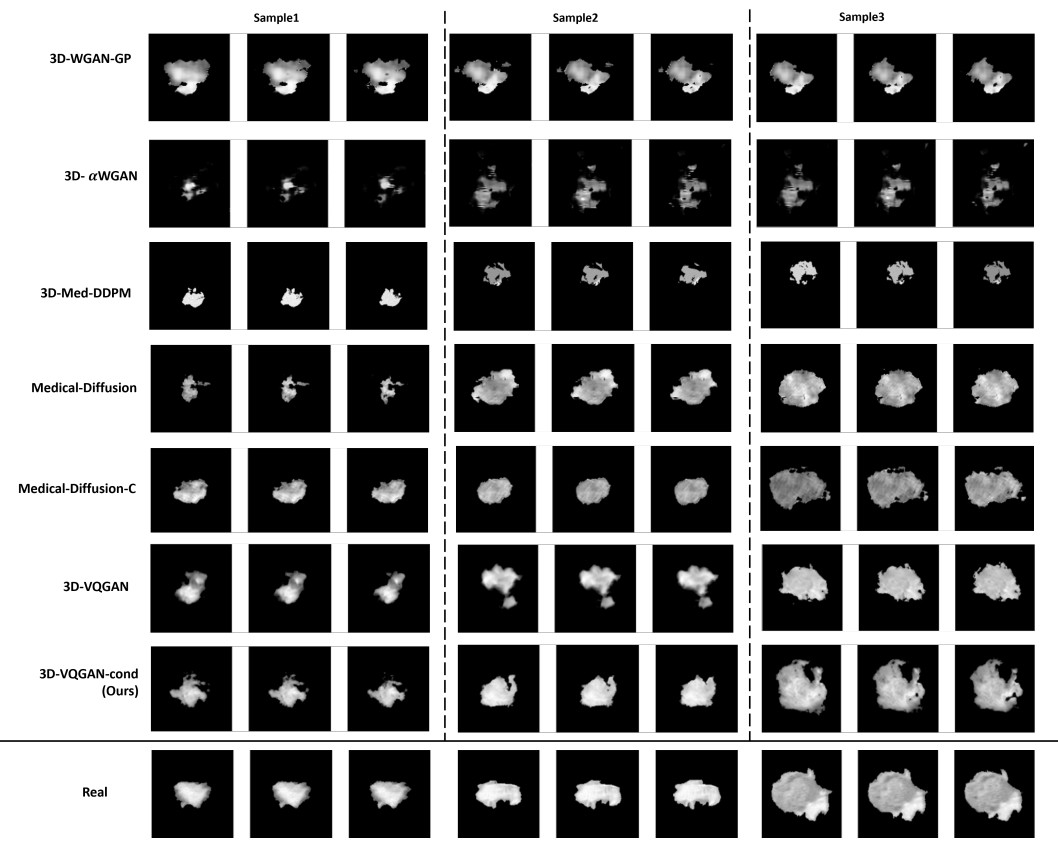

Figure 4: Additional generated samples of HGG data. We show the center three consecutive slices in the Axial plane for each ROI sample. Zoom in for a better view.

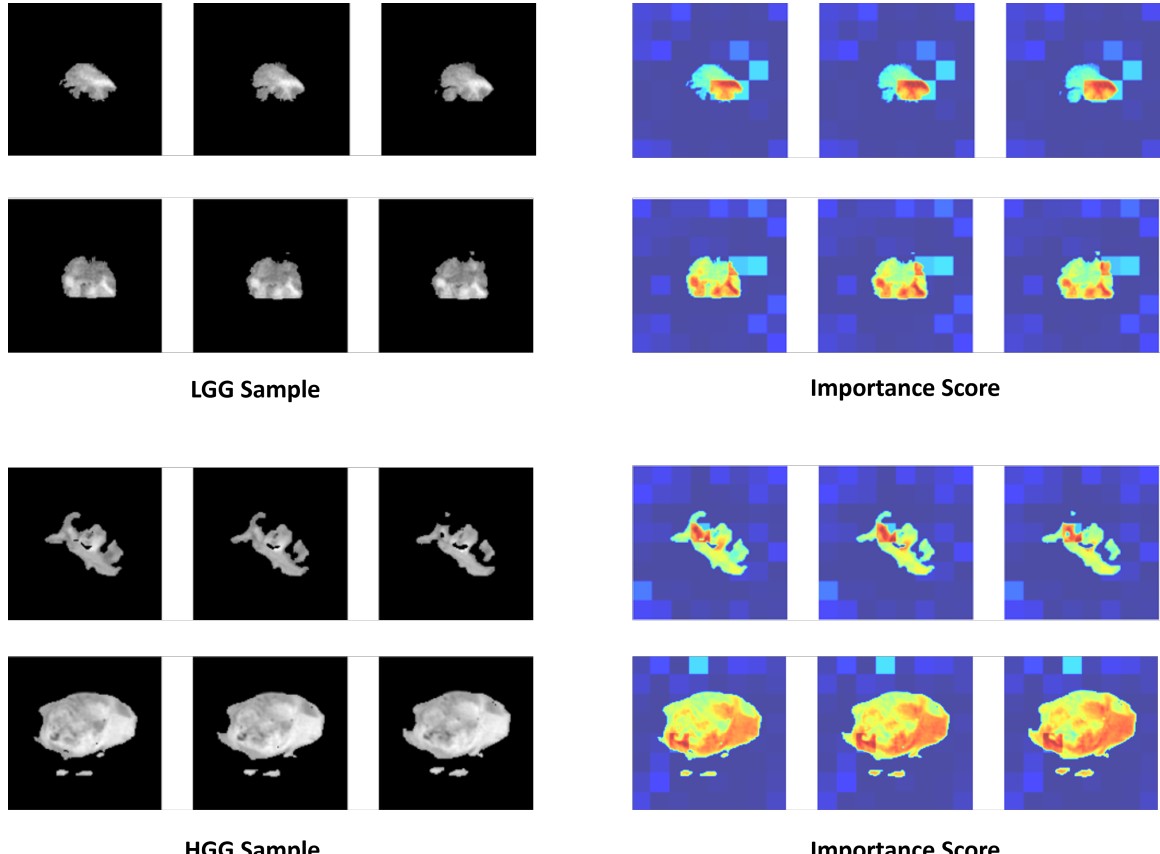

Figure 5: Visualization of importance score map. We show the center three consecutive slices in the Axial plane and its corresponding importance map for two randomly selected LGG and HGG samples.

## Appendix D. More on Classification Results

### D.1. Full Results

In this section, we provide more details on classification performance, this includes the additional report of precision and recall scores for all experiments we performed. Results for **Experiment (1)** is shown in Table 5 and for **Experiment (2)** is shown in Table 6.

Table 5: Detailed classification results for Experiment (1). We run all for three trials and report as mean±standard deviation. **Bold** values represent the best results, and Underline values represent the second-best results.

|  | AUC | F1-Score | Accuracy | Precision | Recall |
|---|---|---|---|---|---|
| Trad. Aug. | 0.66±0.03 | 0.63±0.03 | 0.59±0.03 | 0.59±0.07 | 0.72±0.14 |
| 3D-WGAN-GP | 0.64±0.08 | 0.62±0.05 | 0.56±0.01 | 0.61±0.05 | 0.65±0.24 |
| 3D-$\alpha$WGAN | 0.70±0.09 | 0.59±0.09 | 0.58±0.06 | 0.63±0.12 | 0.67±0.28 |
| 3D-Med-DDPM | 0.70±0.03 | 0.64±0.01 | 0.61±0.06 | 0.62±0.09 | 0.71±0.13 |
| Medical Diffusion | 0.71±0.09 | 0.62±0.08 | 0.59±0.02 | 0.59±0.04 | 0.71±0.20 |
| Medical Diffusion-C | 0.66±0.03 | 0.65±0.02 | 0.58±0.08 | 0.52±0.02 | 0.76±0.11 |
| 3D-VQGAN | 0.72±0.03 | 0.67±0.02 | 0.65±0.04 | 0.64±0.08 | 0.73±0.14 |
| 3D-VQGAN-cond (Ours) | **0.77±0.03** | **0.71±0.02** | **0.67±0.04** | **0.65±0.05** | **0.79±0.07** |

Table 6: Detailed classification results for Experiment (2). We run all for three trials and report as mean±standard deviation. **Bold** values represent the best results, and Underline values represent the second-best results.

|  | AUC | F1-Score | Accuracy | Precision | Recall |
|---|---|---|---|---|---|
| 3D-$\alpha$WGAN | 0.71±0.04 | 0.65±0.02 | 0.64±0.06 | 0.65±0.08 | 0.68±0.10 |
| 3D-Med-DDPM | 0.71±0.09 | 0.69±0.02 | 0.65±0.03 | 0.62±0.04 | 0.77±0.04 |
| Medical Diffusion | 0.73±0.04 | 0.67±0.02 | 0.61±0.02 | 0.55±0.04 | 0.84±0.11 |
| Medical Diffusion-C | 0.75±0.03 | 0.68±0.02 | 0.66±0.03 | 0.64±0.04 | 0.73±0.04 |
| 3D-VQGAN | 0.78±0.04 | 0.71±0.06 | 0.70±0.06 | **0.69±0.06** | 0.72±0.07 |
| 3D-VQGAN-cond (Ours) | **0.80±0.02** | **0.74±0.02** | **0.70±0.01** | 0.66±0.05 | **0.87±0.13** |

### D.2. Comparison with Transfer Learning

We also compare our proposed method with the traditional transfer learning approach. We used the MedicalNet (Chen et al., 2019) pre-trained weights in this work. The model was originally designed for 3D medical image segmentation. Therefore, we adapted the model for our classification task by replacing its segmentation head with a classification head. This new head, implemented as a two-layer MLP, takes the latent representations as input and produces class logits as output for the two tumor types. Due to our computational

limitations, we opted for the ResNet-34 backbone instead. During transfer learning, we froze the feature extractor and only trained the new classification head. The results are included in Table 7. Our method outperforms MedicalNet by a significant margin in all metrics, demonstrating the effectiveness of our proposed approach. Although there is a difference with the model backbone we used, we hypothesize that this change would not dramatically alter the results. This hypothesis is based on the datasets used to pre-train the model were whole 3D volumes of various organs, including but not limited to the brain. Moreover, the substantial difference between whole 3D volumes and Regions of Interest (ROIs) can also affect the results.

Table 7: Comparison between our method's best performance and transfer learning approach. **Bold** values represent the best results. *: results computed using ResNet-34 backbone instead of ResNet-50.

|  | AUC | F1-Score | Accuracy | Precision | Recall |
|---|---|---|---|---|---|
| MedicalNet* | 0.61±0.07 | 0.55±0.03 | 0.55±0.05 | 0.58±0.04 | 0.63±0.05 |
| 3D-VQGAN-cond (Ours) | **0.80±0.02** | **0.74±0.02** | **0.70±0.01** | **0.66±0.05** | **0.87±0.13** |

