# OpenReview forum: "Conditional Generation of 3D Brain Tumor Regions via VQGAN and Temporal-Agnostic Masked Transformer"
_MIDL.io/2024/Conference — MIDL 2024 Poster_

### Official Review · Reviewer_h3no · 2024-02-26

**Confidence:** 3
**Preliminary Rating:** 4
**Recommendation:** Poster
**Final Rating:** 4

**Summary:**

This work aimed at 3D brain tumor ROI generation task as well as downstream classification task. This work adopts VQVAE and VQGAN structure to learn a discretized latent space for brain tumor ROI generation. The main contribution and novelty of this work could be summarized as 1. converting the latent feature into importance map by introducing an auxilary classification module and loss; 2. using a class-conditioned transformer and a proposed temporal-agnostic hybrid token masking method to improve the performance of the transformer. The proposed approach is demonstrated on a single dataset with LGG and HGG data. The results show the success of the proposed method in both quality of generated data and downstream classification accuracy.

**Strengths:**

1. This work uses the structure of 3D-VQVAE and a transformer block to learn the code book as proposed in previous studies. The authors proposed a novel class condition and temporal-spatial masking strategy.
2. The paper presents a comprehensive set of experiments with well-designed baselines.
3. This paper is well written and well organized.

**Weaknesses:**

1. The size of the dataset is not that large. Also the dataset is not balanced for this two classes. I wonder if the authors have done anything to address that issue.
2. I didn't see an ablation study in terms of the masking strategy. I wonder if the authors have done that or not.
3. The comparisons with diffusion model is not that fair. It's very easy and straight forward to add class condition to diffusion models. In this work, the diffusion model is trained separately.

**Detailed Comments:**

1. Is the proposed masking method working better? Does the way of choosing the number of neighbor matter?

2. The results of generated ROI is not that straightforward. It's hard for me to tell which generated ROI is better. Please highlight the region that you think is better.

3. It would be better to add more ablation studies as mentioned in the Questions to Address section below.

**Justification Of Final Rating:**

I think the authors have successfully addressed my concerns and I'm satisfied with the additional experiments. They added more experiments on the ablation studies of different ratio of masking. Thank the authors for their efforts and this great work.

**Justification Of The Preliminary Rating:**

The overall quality of the paper is high. The idea is novel where the authors proposed a conditioned transformer and masking strategy. The experiments show the effectiveness of the proposed method. Based on the merits of the paper, I would like to weakly accept the paper.

**Questions To Address In The Rebuttal:**

Similar to detailed comments.

1. I would like to see some explanations from the authors about the proposed masking strategy with ablation studies. Is it working? How does it work? Also whether the choice of the number of neighbors matters?

2. I would like to see some comparisons with diffusion models with condition.

3. I wonder if the author could explore their proposed method on more complicated data, e.g., larger ROI, more classes instead of 2.

**Special Issue:**

No

---

> ### Author Response · Authors · 2024-03-16
> **Author Response to Reviewer h3no**
>
> We thank the reviewer for the constructive comments.
>
> **Q1: Ablations on the proposed masking strategy**
>
> We have conducted two ablation studies. We focused on the top-k ratio we defined in Section 2.1 (ratio of important tokens of all discrete semantic tokens obtained from encoded feature maps), and also the number of neighboring tokens you have suggested. We claimed that the top-k ratio controls the trade-off between the diversity of generated images (measured by MS-SSIM score) and their quality (measured by MMD score). We have conducted experiments on k=0, 25, and 50% and reported two scores mentioned above on the generated samples. Notice that when k=0, we treat every token as unimportant, thus the proposed masking strategy degrades to random masking. Results show that when k=0%, we have a slightly lower MMD but much higher MS-SSIM, when k=25%, we have a slightly higher MMD than k=0% but much lower MS-SSIM score, when k=50%, we have a much higher MMD and almost the same MS-SSIM with k=25%. Hence, to balance this trade-off, we select k=25% in this study. Next, we conducted ablations on the number of neighboring tokens to be masked. We masked only one neighbor token in either spatial or temporal dimension in our work, so the ablation is done by masking two neighboring tokens around the important one. We observed a dramatic increase in the MMD score for double-sided masking. Please refer to Appendix B for full details.
>
> **Q2: Diffusion model with conditions**
>
> We have re-implemented a class-conditioned version of Medical Diffusion (termed Medical Diffusion-C), since it is the most similar architecture compared to ours (VQGAN+Diffusion vs. VQGAN+Transformer) and is computationally friendly. Although Medical Diffusion-C outperforms in FID score based on generated samples, ours is better in other quantitative metrics as well as classification results. Please see Table 1 and 2, Figure 2,3, and 4, and Appendix D for details.
>
> **Q3: Hard to tell which generated ROI is better**
>
> We agree with the reviewer. As a non-radiologist, it is challenging to tell the difference between generated ROIs from a clinical perspective. But here, we provide some of our own thoughts when we look at these ROIs in Figure 3 and 4. All GAN-based baselines produce images with noise and blurry edges, and the image quality is low. For the 3D-Med-DDPM, the intensity range in the generated samples seems to mismatch the real samples, and it looks unreal compared with the real ROIs. For Medical Diffusion/-C, the generated images suffer from minor checkerboard artifacts (visible when zoomed in). 3D-VQGAN sometimes produce blurry images (sample 2 in HGG), but overall, the generated images are smooth and do not have any checkerboard artifacts or noises. Finally, for our method, the images exhibit high-resolution with no noise, no blurry edges, and no checkerboard artifacts, the contrast inside the generated ROIs looks similar to the real ROIs. Therefore, we think our proposed method produces the best ROIs in terms of visual quality. We have also included this in Appendix C, highlighted in red.
>
> **Q4: Proposed method on more complicated datasets**
>
> We agree with the reviewer that it would make our proposed method stronger if we could evaluate it on different datasets. However, considering the costs and efforts to get more complex datasets, we think this would be a great line of future work to work on. We envision that our method can adopt to any other tumor types as well.

---

> ### Author Response · Authors · 2024-03-26
> **Looking forward to further feedback**
>
> Dear Reviewer h3no,
>
> Thank you again for your valuable comments and suggestions, which are very helpful to us. We have posted responses above to your concerns.
>
> We understand that this is quite a busy period, so we sincerely appreciate it if you could take some time to return further feedback on whether our responses address your concerns. If there are any other comments, we will try our best to answer them.
>
> Best,
>
> The Authors

---

> > ### Comment · Reviewer_h3no · 2024-03-26
> >
> > Congrats on your successful revisions! My concerns are addressed and thank you for your efforts on the paper and revision.

---

> > > ### Author Response · Authors · 2024-03-26
> > > **Thank you**
> > >
> > > Thank you for your feedback! We are happy to hear that we were able to address all of your concerns.

---

### Official Review · Reviewer_jbth · 2024-02-27

**Confidence:** 3
**Preliminary Rating:** 2
**Final Rating:** 4

**Summary:**

This paper proposes to use 3D VQ-GAN to generate 3D brain tumour ROIs, instead of the entire brain, which can stabilises the training of GANs by reduce data dimension. Evaluation is performed by evaluating the quality of synthetic brain tumour ROIs and also by downstream task training on synthetic data.

**Strengths:**

- The proposed method focus on generating 3D brain tumour ROIs with VQ-GAN and a transformer. The generated ROIs look plausible and the quantitative numbers look fine.
- The trained model can produce ROIs for HGG or LGG conditioned on input variable.

**Weaknesses:**

- Evaluation is only on one dataset with 259 HGG patients and 76 LGG patients.
- Technical novelty/contribution is limited. Despite it focused on ROIs, the differences between this paper and previous works on 3D brains are minimal.
- Brain tumour can not only affect the ROI, but also affect the other parts of brains by causing deformations. Maybe have some discussion on it?
- ROIs are center-cropped to "a target size of 128 × 128 × 128”. As the authors mentioned, they focused on ROIs because “GANs can be unstable, produce blurry images, and encounter mode collapse problems”. However, 128*128*128 is still large, and since authors used GANs in this paper, how did they ensure that the unstable training did not occur?
- What if we reshaped a whole 3d brain to 128*128*128, and generate the downsampled version brains?

**Detailed Comments:**

- Figure 2 and 3 are not informative. The results figures are small. And I found difficult to tell differences between the proposed method and baseline.
- Some literature is missing.
Sun L, Wang J, Huang Y, et al. An adversarial learning approach to medical image synthesis for lesion detection[J]. IEEE journal of biomedical and health informatics, 2020, 24(8): 2303-2314.

Sanchez P, Kascenas A, Liu X, et al. What is healthy? generative counterfactual diffusion for lesion localization[C]//MICCAI Workshop on Deep Generative Models. Cham: Springer Nature Switzerland, 2022: 34-44.

Xia T, Chartsias A, Tsaftaris S A. Pseudo-healthy synthesis with pathology disentanglement and adversarial learning[J]. Medical Image Analysis, 2020, 64: 101719

**Justification Of Final Rating:**

This paper proposes to use 3D VQ-GAN to generate 3D brain tumour ROIs conditioned on labels of LGG and HGG. I found the technical novelty is limited, and I do not follow the advantages of generating ROIs rather than whole brains. The experiments only focus on comparing different generative models for ROIs, but would be better to see experiments comparing generating ROIs and whole brains. The claimed benefit of focusing on ROIs is because GANs are unstable, but the size for ROIs is set as 128128128, which is large and I am not sure how this is going to help stabilise GAN training.

====update======
The authors’ responses have addressed most of my concerns. I would like to update rating accordingly.

**Justification Of The Preliminary Rating:**

This paper proposes to use 3D VQ-GAN to generate 3D brain tumour ROIs conditioned on labels of LGG and HGG. I found the technical novelty is limited, and I do not follow the advantages of generating ROIs rather than whole brains. The experiments only focus on comparing different generative models for ROIs, but would be better to see experiments comparing generating ROIs and whole brains. The claimed benefit of focusing on ROIs is because GANs are unstable, but the size for ROIs is set as 128*128*128, which is large and I am not sure how this is going to help stabilise GAN training.

**Questions To Address In The Rebuttal:**

- Brain tumour can not only affect the ROI, but also affect the other parts of brains by causing deformations. Maybe have some discussion on it?
- ROIs are center-cropped to "a target size of 128 × 128 × 128”. As the authors mentioned, they focused on ROIs because “GANs can be unstable, produce blurry images, and encounter mode collapse problems”. However, 128*128*128 is still large, and since authors used GANs in this paper, how did they ensure that the unstable training did not occur?
- What if we reshaped a whole 3d brain to 128*128*128, and generate the downsampled version brains?
- Improve Figure 2 and 3, I think might be worth to have similar real images by side to better demonstrate image quality.

---

> ### Author Response · Authors · 2024-03-16
> **Author Response to Reviewer jbth**
>
> Thank you for your review and insightful comments.
>
> **Q1: Discussion on why ROI-based classification for the brain tumor, and missing literature**
>
> This is a great question. In fact, the brain tumor types diagnosis/classification based on MRI are ROI-based in recent studies [1,2,3]. We are classifying tumor types in this work (HGG vs. LGG), rather than classifying whether the tumor is present in the brain or not. In this case, the key difference between **tumor pathologies (types)** would be based on the tumor area (ROI), and hence, the information outside the ROI may negatively affect the results. We have added this to Section 1 (page 2), highlighted in red. We have also added you suggested literature to Section 1 (page 2).
>
> **Q2: GAN stability for 128x128x128 images**
>
> We would like to re-emphasize on the VQGAN architecture that we used in this work. The goal of VQGAN is to learn a CNN-based autoencoder, and a codebook to quantize the latent feature maps to discrete semantic tokens. The overall workflow of VQGAN is first to encode the image into latent features, perform quantization and obtain quantized features from the codebook, and decode them back to the original image. To supervise the training of VQGAN, we follow the original paper [4] to use an adversarial training approach (GAN) to differentiate between real and reconstructed images, in addition to conventional reconstruction objectives such as L1 loss, perceptual loss, etc. Therefore, in our case, the GAN is only used to force the reconstructed image from the entire workflow to be the same as the original image in an adversarial manner. Unlike other conventional GAN models, **we are not using GAN to generate any new images**. In Section 1, we refer GANs that are unstable and produce blurry images as those models that generate images from random noise.  In our work, the image generation happens in our proposed class-conditional masked transformer model.
>
> **Q3: Re-emphasize novelty**
>
> We want to reemphasize our novelty here; we agree with the review that this paper is based on some existing previous works on image generation. However, there are substantial differences with previous publications.
>
> 1. To the best of our knowledge, this is the first attempt to use VQGAN and transformer to generate high-resolution brain tumor types (HGG and LGG) in a class-conditioned manner.
>
> 2. The proposed masking strategy introduces a novel approach by considering the importance of each token during the masking process, contrasting with conventional random masking methods. The proposed strategy prioritizes tokens based on their importance scores and we perform block-wise masking around them to improve the model learnability and robustness. We have shown the effectiveness of this approach by conducting ablation studies in Appendix B.
>
> 3. To learn the importance score, we proposed a classifier-guidance approach by using an auxiliary classifier to guide the learning process. Although classifier-guidance is not new, it is the first time this approach applied to the medical image data to effectively learn the importance of each region feature.
>
> **Q4: Reshape whole 3D brain to 128x128x128 and generate the downsampled version brains**
>
> As we stated in the previous question, the current effort focuses on ROI-based classification of brain tumor pathology. Hence in this work we continue this line of work and only use ROI instead of the whole brain. We agree with the reviewer that it is possible to work with the whole brain. However, in that case, we need either to generate synthetic whole-brain images with the corresponding segmentations simultaneously, or we need to use a subsequent segmentation model to get the ROI, this is somehow out of the scope of our study. We think the proposed approach by the reviewer is a great line of future work to continue.
>
> **Q5: Include real ROIs in figures**
>
> It is quite hard to enlarge figures in the main text due to the page limit, but we have included the real ROIs in Figure 3 and 4 and enlarged them in Appendix C for better visualization and comparison.

---

> > ### Author Response · Authors · 2024-03-16
> > **References**
> >
> > [1]. Sajjad, M., Khan, S., Muhammad, K., Wu, W., Ullah, A., & Baik, S. W. (2019). Multi-grade brain tumor classification using deep CNN with extensive data augmentation. Journal of computational science, 30, 174-182.
> >
> > [2]. Mzoughi, H., Njeh, I., Wali, A., Slima, M. B., BenHamida, A., Mhiri, C., & Mahfoudhe, K. B. (2020). Deep multi-scale 3D convolutional neural network (CNN) for MRI gliomas brain tumor classification. Journal of Digital Imaging, 33, 903-915.
> >
> > [3]. Srinivasan, S., Bai, P. S. M., Mathivanan, S. K., Muthukumaran, V., Babu, J. C., & Vilcekova, L. (2023). Grade Classification of Tumors from Brain Magnetic Resonance Images Using a Deep Learning Technique. Diagnostics, 13(6), 1153.
> >
> > [4]. Esser, P., Rombach, R., & Ommer, B. (2021). Taming transformers for high-resolution image synthesis. In Proceedings of the IEEE/CVF conference on computer vision and pattern recognition (pp. 12873-12883).

---

> > > ### Comment · Reviewer_jbth · 2024-03-20
> > > **Update of comments**
> > >
> > > Thank the authors for the response. It addresses most of my concerns and I update my rating accordingly.

---

> > > > ### Author Response · Authors · 2024-03-21
> > > > **Clarification Regarding the Updated Rating**
> > > >
> > > > We would like to express our gratitude again for your insightful comments and we are happy to see that our responses address most of your concerns and thanks for updating the rating.
> > > >
> > > > However, upon checking the author console, we could not see the change in the rating. Could you please kindly double-check that the rating has been successfully updated? Thank you once again for your review!

---

> > > > > ### Comment · Reviewer_jbth · 2024-03-21
> > > > > **Reply to Clarification Regarding the Updated Rating**
> > > > >
> > > > > Thanks. I have double checked it has been successfully updated. Somehow it is only visible to ACs, Reviewers and PCs. I tried to set it visible to Authors but couldn’t make it. Anyway, Rest assured that the rating is updated.

---

> > > > > > ### Author Response · Authors · 2024-03-21
> > > > > > **Thank you**
> > > > > >
> > > > > > Thank you for letting us know, we appreciate it.

---

### Official Review · Reviewer_Y4ZB · 2024-03-05

**Confidence:** 4
**Preliminary Rating:** 4
**Recommendation:** Poster
**Final Rating:** 4

**Summary:**

The authors presented a generative GAN-based method for generating synthesized brain tumor ROIs, and verified its added benefit for improving the tumor classification performance.

**Strengths:**

The application of GAN-based tumor ROI generation and the temporal-agnostic hybrid masking strategy seem to offer interesting insights to the community. The findings are well supported by experimental results.

**Weaknesses:**

The selection of modules and methods is less motivated and verified. The code seems not to be available, making it hard for other researchers to reproduce and compare. It is strongly recommended to release the code for the convenience of the research community.

**Detailed Comments:**

- The word "class-conditioned" or "conditional" is a bit unclear and perhaps misleading. It is not fully clear how the network is conditioned on the class label to generate different ROIs. If I am not mistaken, it is incorporated via the MLP classifier. However, it is not guaranteed that the label would be a forced condition; it is rather a guidance. I would suggest using "guided" if the method could not use labels as an absolute conditional input.
- The classifier may make mistakes in classifying the labels. If case the classification is wrong, would it mislead the generated importance score map? It would be valuable to evaluate the impact of incorrect classification on the importance score map.
- The authors mentioned that one line of related work is transfer learning. It would be interesting to compare this method with the transfer learning strategy.
- Please add statistical calculations in Table 2. Accordingly in the conclusion, please clarify the statistical significance in the statement of "our method yields the best ..."

**Justification Of Final Rating:**

Heartfelt thanks to the authors for carefully addressing my questions and concerns. The authors present an innovative generative GAN-based methodology and assess its added value on performance with experiment results; this could offer valuable insight to the research community. I would still strongly recommend the authors make the code publicly available, with good documentation.

**Justification Of The Preliminary Rating:**

The proposed methodology seems to be insightful. However, the authors need to be better explain how the conditions are forced in the unified model, and what the impacts are in case of wrong classifications.

**Questions To Address In The Rebuttal:**

see detailed comments.

**Special Issue:**

No

---

> ### Author Response · Authors · 2024-03-16
> **Author Response to Reviewer Y4ZB**
>
> We thank the reviewer for the detailed feedback and insightful comments.
>
> **Q1: Class-conditioned vs. Class-guided**
>
> When we train the transformer, we prepend a class token to the discrete semantic representation for each sample we obtain from the codebook. The transformer will learn the relationship between the semantic tokens given the information of the class label during training. When we generate the synthetic image, we only provide a class label (0 or 1) to the transformer, and let it autoregressively generate the next token to obtain a full sequence of tokens, which are then decoded back to the image. Hence, since we use a single label describing the overall image class and use it as a condition to generate the synthetic image, we believe it should be class-conditioned, following the terminology from previous publications [1,2]. We have changed ‘conditional 3D-VQGAN’ to ‘3D-VQGAN’ and ‘Conditional Masked Transformer’ to ‘class-conditional masked transformer’ to avoid any confusion (Figure 1, page 3).
>
> **Q2: Misclassification in importance score map**
>
> The goal of our proposed auxiliary MLP classifier is to learn which region in the latent feature maps is important and assign a weight (score) with respect to the tumor type (HGG/LGG). If the region can best reflect the tumor or can be used to differentiate tumor types, it should receive a higher weight. In the second stage, when we train our transformer model, we use the importance score map to get the important/unimportant regions in the feature maps and perform masking for each sample. The importance score map is not used in either ROI generation or the downstream classification task. We agree that there is a chance that the misclassification problem would occur, but this only affects the weight of feature regions, and whether or not they are important, and the way they are masked. Since our masking strategy is hybrid, we argue that this hybrid masking is relatively robust to this problem.
>
> **Q3: Compare with transfer learning and Add p-value to the results**
>
> We have conducted the independent-sample two-sided t-test on the AUC and F1-score of our proposed method and compared them with the previous SOTA (3D-VQGAN). Both p-values for AUC and F1-score are statistically significant (p < 0.05 in all cases). For the transfer learning baseline, we used MedicalNet [3] pre-trained weights with their ResNet-34 backbone instead of ResNet-50 due to our computational limitation. We froze the feature extractor and only trained the classification head. Results and details can be found in Appendix D.2.
>
> References:
>
> [1]. Esser, P., Rombach, R., & Ommer, B. (2021). Taming transformers for high-resolution image synthesis. In Proceedings of the IEEE/CVF conference on computer vision and pattern recognition (pp. 12873-12883).
>
> [2]. Yu, J., Li, X., Koh, J. Y., Zhang, H., Pang, R., Qin, J., ... & Wu, Y. (2021). Vector-quantized image modeling with improved vqgan. arXiv preprint arXiv:2110.04627.
>
> [3]. Chen, S., Ma, K., & Zheng, Y. (2019). Med3d: Transfer learning for 3d medical image analysis. arXiv preprint arXiv:1904.00625.

---

> ### Author Response · Authors · 2024-03-26
> **Looking forward to further feedback**
>
> Dear Reviewer Y4ZB,
>
> Thank you again for your valuable comments and suggestions, which are very helpful to us. We have posted responses above to your concerns.
>
> We understand that this is quite a busy period, so we sincerely appreciate it if you could take some time to return further feedback on whether our responses address your concerns. If there are any other comments, we will try our best to answer them.
>
> Best,
>
> The Authors

---

> > ### Comment · Reviewer_Y4ZB · 2024-03-26
> >
> > Heartfelt thanks to the authors for addressing my concerns. I have no more questions.

---

> > > ### Author Response · Authors · 2024-03-26
> > > **Thank you**
> > >
> > > Thank you for your feedback! We are happy to hear that we were able to address all of your concerns.

---

### Author Response · Authors · 2024-03-16
**Summary of Changes**

We would like to thank all the reviewers for their helpful and constructive feedback, and insightful suggestions. We have addressed these comments and below, we summarize the major changes to the paper.

1.	We added the p-value of our proposed method compared with the previous SOTA (3D-VQGAN) in Table 1, and included the results for class-conditional Medical Diffusion as our ‘class-conditional’ baseline (Reviewer Y4ZB, h3no).

2.	We compared our proposed method with a transfer learning-based approach, (MedicalNet [1]), and reported in classification results in Appendix D.2. (Reviewer Y4ZB).

3.	We included some discussions in ROI-based vs. whole-brain-based classification, and added more references in Section 1, page 2 (Reviewer jbth).

4.	We conducted extensive ablation studies and reported the results in Appendix B (Reviewer h3no).

5.	We included the generated samples of class-conditional Medical Diffusion in Figure 2,3, and 4. We also enlarged Figure 3 and 4 in Appendix C and included real ROIs for better visualization and comparison. We provided a brief analysis of the image quality of the generated images for all methods in Appendix C, highlighted in red (Reviewer jbth, h3no).

6.	Due to the page limit, we decided to move some implementation details regarding the importance score map from the main text to Appendix A.1.

We are working on the code and will release them in the near future.

Reference:

[1]. Chen, S., Ma, K., & Zheng, Y. (2019). Med3d: Transfer learning for 3d medical image analysis. arXiv preprint arXiv:1904.00625.

---

### Meta-Review · Area_Chair_sVFi · 2024-03-29

**Recommendation:** Accept (Poster)
**Confidence:** 5

**Metareview:**

After a busy and productive discussion between authors and reviewers (including new results), the paper is very much improved and all authors agree that the MIDL community will see the value on their method.

---

### Decision · Program_Chairs · 2024-04-05

Accept (Poster)